# Canadian Arctic Contaminants and Their Effects on the Maternal Brain and Behaviour: A Scoping Review of the Animal Literature

**DOI:** 10.3390/ijerph17030926

**Published:** 2020-02-02

**Authors:** Claire Fong-McMaster, Sandra Konji, Amanda Nitschke, Anne TM Konkle

**Affiliations:** 1Interdisciplinary School of Health Sciences, University of Ottawa, Ottawa, ON K1N 6N5, Canada; cfong006@uottawa.ca (C.F.-M.); sandrakonji@gmail.com (S.K.); anits104@uottawa.ca (A.N.); 2School of Psychology, University of Ottawa, Ottawa, ON K1N 6N5, Canada; 3University of Ottawa Brain and Mind Research Institute, University of Ottawa, Ottawa, ON K1H 8M5, Canada

**Keywords:** maternal behaviour, maternal brain, pregnancy, postpartum, methylmercury, polychlorinated biphenyls, organochlorine pesticides, persistent organic pollutants

## Abstract

**Background:** Environmental toxicants such as methylmercury, polychlorinated biphenyls, and organochlorine pesticides are potentially harmful pollutants present in contaminated food, soil, air, and water. Exposure to these ecologically relevant toxicants is prominent in Northern Canadian populations. Previous work focused on toxicant exposure during pregnancy as a threat to fetal neurodevelopment. However, little is known about the individual and combined effects of these toxicants on maternal health during pregnancy and post-partum. **Methods:** A scoping review was conducted to synthesize the current knowledge regarding individual and combined effects of methylmercury, polychlorinated biphenyls, and organochlorine pesticides on maternal behaviour and the maternal brain. Relevant studies were identified through the PubMed, Embase, and Toxline databases. Literature involving animal models and one human cohort were included in the review. **Results:** Research findings indicate that exposures to these environmental toxicants are associated with neurochemical changes in rodent models. Animal models provided the majority of information on toxicant-induced alterations in maternal care behaviours. Molecular and hormonal changes hypothesized to underlie these alterations were also addressed, although studies assessing toxicant co-exposure were limited. **Conclusion:** This review speaks to the limited knowledge regarding effects of these persistent organic pollutants on the maternal brain and related behavioural outcomes. Further research is required to better comprehend any such effects on maternal brain and behaviour, as maternal care is an important contributor to offspring neurodevelopment.

## 1. Introduction

Environmental pollution is a global problem associated with many adverse health outcomes [1,2,3]. Organic and inorganic pollutants contaminate soil, air, water, and food in many urban and rural communities [4,5,6,7]. Of particular concern are polychlorinated biphenyls (PCBs) and organochlorine pesticides (OCPs), which are two classes of industrial chemicals that persist in the environment due to their stable chemical structure and long half-life [8,9]. Mercury is another potent environmental contaminant that exerts toxic effects on a variety of vital organs [10]. Due to these concerns, international agreements have been established to limit the production and usage of these chemicals [11,12].

### 1.1. Methylmercury 

Mercury is naturally found in three predominant forms: elemental, organic, and inorganic mercury [13]. Organic methylmercury (MeHg) is the most toxic of all forms. Humans are primarily exposed through consumption of contaminated fish and marine mammals, particularly large long-lived predators at the top of the food web [3,14]. Ingested MeHg is absorbed via the digestive tract and can accumulate in the brain throughout life [15]. As a result of increasing fish consumption, chronic low-level dietary intake of MeHg has become more prevalent and as such may pose a significant toxicological risk [16]. 

### 1.2. Polychlorinated Biphenyls

PCBs are contaminants that were widely used in industrial machinery such as transformers, capacitors, hydraulic fluids, and lubricants until the 1970s [17]. This class of contaminants includes 209 chemical congeners that differ in the extent and pattern of chlorination of the biphenyl structure [18]. An increase in chlorine substitutions further decreases the aqueous solubility and degradability of PCB compounds. Thus, the structural composition has direct effects on the environmental fate and toxicity of each congener [19]. 

Although most industrialized countries have banned their production and use, PCBs remain widely dispersed in the environment due to their bioaccumulation in food webs [20]. Atmospheric deposition of PCBs is higher in the Arctic region compared to lower latitudes [21]. Generally, food samples from the northern hemisphere contain greater PCBs compared to the southern hemisphere [22]. Exposure to PCBs most often results from ingestion of contaminated foods such as fish, meat, and milk [23]. 

### 1.3. Organochlorine Pesticides

OCPs are a group of chlorinated compounds that were extensively used in the agricultural industry between the 1950s and 1970s [24,25]. These are a large group of compounds that consist of aliphatic and aromatic cyclical structures with multiple chlorine substitutions [25]. OCPs persist in the environment due to their stability and slow biodegradation [26]. OCPs such as hexachlorocyclohexane and endosulfan are found in high concentrations in Asia, while varied levels are observed across North America [27]. Similar to PCBs, air–seawater monitoring suggests the net deposition of OCPs in the Arctic region [28]. 

Since the 1970s, widespread use of OCPs has been drastically restricted because of concerns about their environmental persistence, bioaccumulation, and potential to cause adverse effects in humans [29]. Dietary consumption of fatty foods such as fish is now the main source of exposure [24] due to their bioaccumulation in these animals.

### 1.4. Contaminant Mixtures

Complex combinations of environmental toxicants are prevalent in the Canadian Arctic. As a result, individuals are typically exposed to multiple pollutants over time [30]. The Canadian Arctic region in particular is vulnerable to the aforementioned contaminants due to long-range transport and oceanic currents, which drive their accumulation in this region’s environment and biota [31]. The lipophilic nature and persistence of MeHg, PCBs, and OCPs allow for their bioaccumulation in high trophic level species including whales, walruses, seals, and fish [30,32,33,34,35]. Northern Aboriginal populations, over 56,000 individuals from Labrador, Northwest Territories, Nunavik, Nunavut, and Yukon, have a heavy dietary reliance on these food sources [36,37]. As such, they are exposed to high levels of complex toxicant combinations. Biomonitoring studies have indicated that contaminant burden is higher in populations that consume large amounts of traditional foods from the marine environment than in those that do not [38,39,40]. For example, PCB and OCP levels measured in breast milk of Nunavik Inuit mothers were nearly ten-fold higher than those found in Southern Canada mothers [39,41,42,43]. Similarly, almost 25% of Aboriginal peoples of Northern and Eastern Inuit communities had MeHg levels above 20 ppm in blood (6 ppm in hair samples), the acceptable limit determined by the World Health Organization at the time of this study [44].

To investigate the possible health effects of simultaneous exposure to multiple pollutants in the Canadian Arctic communities, the Northern Contaminant Program was developed in 1991 under the Ministry of Indigenous and Northern Affairs Canada [30]. Work under this program sought to better understand the consequences of exposure to a mixture of these toxicants. As such, Health Canada developed the Northern Contaminant Mixture (NCM) to be used for testing in animal models [45]. The NCM was formulated to comprise the 27 most abundant environmental contaminants: 14 PCB congeners, 12 OCPs and MeHg, which have been detected in the blood profiles of 159 mothers residing in the Canadian Arctic [45,46].

### 1.5. Maternal Brain and Behaviour

Pregnancy modifies physiological and neuroendocrine processes, and the resulting behavioural adaptations allow the postpartum female to effectively care for her young [47]. An increased ratio of estradiol to progesterone and increased prolactin and oxytocin are hormonal events associated with late pregnancy and parturition. These parameters increase in rats to provide initial activation of the maternal neural circuitry and maternal behaviour [48,49]. However, this hormonal influence is transient, which is why sensory experiences acquired through mother–pup interactions are essential for the continuance of maternal responsiveness [50,51]. 

Many researchers have explored the impact of the abovementioned pollutants on the neurological health of offspring [52,53,54,55]. Considering the sensitive period of in utero development, studies have focused on developmental outcomes following toxicant exposure. Multiple epidemiological and experimental studies have identified damaging effects of prenatal toxicant exposure leading to cognitive dysfunction and behavioural alterations [56]. Conversely, little research has sought to explore contaminant effects on mothers during pregnancy and post-partum. Given the mother’s importance in caring for offspring once they are born, any changes in brain plasticity during this period may influence the quality of care she provides to her offspring. The quality of maternal care is paramount during offspring neurodevelopment [57,58,59]. Thus, this paper aims to provide a scoping review of the literature in an attempt to better understand the effects of maternal exposure to MeHg, OCPs, or PCBs on maternal health and behaviour.

## 2. Materials and Methods

A scoping review was conducted to explore the literature in this growing research field. This study was conducted according to the five-stage scoping review framework described by Arksey and O’Malley [60].

### 2.1. Research Question

This review was guided by the research question “What are the individual and combined effects of MeHg, PCB, and OCP exposure on maternal behaviour and the maternal brain?” 

### 2.2. Identifying Relevant Studies

Comprehensive searches of the Medline, Embase, and Toxline electronic databases were conducted between July 11 and 13 2019. These databases were selected to encompass both disciplines of interest—maternal health and environmental toxicology. Articles published between 2000 and 2019 were considered for this review, as the increase in developmental toxicology studies involving maternal brain and behavioural assessments is relatively recent [61,62]. While there are potential confounding factors of epidemiological data, studies with human cohorts and experimental animal models were included due to the limited number of toxicology studies looking at maternal endpoints. [63]. The search strategy and keywords were developed with the assistance of a Health Sciences librarian from the University of Ottawa. Keywords consisted of terms associated with the maternal exposure period, brain outcomes and toxicants, such as pregnancy, perinatal, maternal, gestation, brain, behaviour, MeHg, PCB, OCP, and northern contaminant mixture. Both keywords and subject headings were used for the Medline and Embase searches. Complete search strategy details are shown in Table 1 and Table 2. All citations were imported to the Covidence reference management software (Veritas Health Innovation, Melbourne, Australia) and duplicate articles were immediately removed.

### 2.3. Selection of Studies

All studies included in the analysis: (1) involved a human cohort or animal model; (2) included direct or indirect exposure to MeHg, PCB congeners, OCPs, or co-exposure to any of the three toxicants; (3) included maternal behavioural or molecular assessments of the maternal brain; (4) were primary source literature. Studies were excluded from the analysis if the full text was unavailable. 

Titles and abstracts were first screened to eliminate articles irrelevant to the research objective. Full-text articles for the remaining studies were reviewed to determine eligibility for inclusion in the scoping review. Articles were screened by one reviewer (C.F.M.), and any uncertainties in study selection were discussed with the principal investigator (A.T.M.K.). 

### 2.4. Charting the Data

Nineteen included studies were reviewed and information from each article was abstracted, including: year of study, toxicant of interest, objective, study design, maternal subjects, sample size, treatment groups, exposure route, exposure period, behavioural findings, and neurochemical findings.

### 2.5. Collating, Summarizing, and Reporting the Results

Data extracted from the full-text review were organized using a Microsoft Excel spreadsheet (Microsoft Corporation, Redmond, WA, USA). Studies included in the analysis are shown in Table 3.

## 3. Results

A total of 1194 studies were identified after de-duplication for possible inclusion in the review. A total of 1138 studies were excluded based on the criteria for inclusion listed in Section 2.3 After title and abstract screening, 56 studies remained that were included in the full text review. Eighteen articles met the criteria for inclusion in the scoping review. One study further analysed original data from a previous study, which was subsequently included. Full details of the scoping review process are displayed in Figure 1. All studies used experimental animal models, except for one human cohort study. Many studies reported both maternal and offspring outcomes. For the purpose of this review, only the maternal findings are presented in Table 3; please refer to this table for additional details regarding each study.

### 3.1. Effects of Methylmercury on the Maternal Brain or Behaviour

#### 3.1.1. Maternal Behaviour

In one study, Weston et al. (2014) used rats to monitor maternal behaviours, including passive nursing, arched back nursing, blanket nursing, pup licking and grooming, pup licking, no contact and no contact resting following the administration of MeHg [64]. Mothers (dams) were separated into treatment groups exposed to 0, 0.5, or 2.5 ppm MeHg drinking water. This was administered two to three weeks prior to breeding until offspring weaning. The authors did not report the amount of water consumed in each experimental group, making it difficult to deduce the actual dose consumed. Changes in maternal behaviour attributed to MeHg exposure alone were limited, with no significant group differences [64]. As such, this study suggests a limited effect of MeHg on maternal rodent behaviour.

In a second rodent study, rat dams were exposed to MeHg at 0 or 0.5 mg/kg body weight/day and retinyl palmitate (Vitamin A), either alone or in combination throughout gestational day (GD) 0 to postnatal day (PND) 21 [65]. Similar to the Weston et al. study, no differences in nursing and pup retrieval behaviours were observed between control and treated groups [65]. However, changes in redox parameters in the maternal brain were observed, as described in Section 3.1.2 below.

Another study using a rodent model investigated different speciations of MeHg and their effects on behaviour [66]. Mouse dams were exposed to 0, 1.5, or 4.5 mg/kg methylmercury chloride (MeHgCl) or methylmercury cysteine (MeHgCys) diets ad libitum from six weeks prior to mating until two weeks following birth [66]. Based on the measured feeding rates, mice in the low dose MeHg groups consumed 223–250 μg/kg, and the high dose consumed 596–629 μg/kg body weight per day of MeHgCys or MeHgCl. The high MeHgCl-exposed group exhibited significantly decreased exploratory behaviour compared to the control group. In the elevated plus maze, this group displayed an increased latency to move from the center section compared to the high MeHgCys diet and control group [66]. The elevated plus maze is widely used to assess anxiety-like behaviour in rodent models [67]. This study suggests that high dose MeHgCl may cause anxiety-like behaviours in rodent dams.

Lastly, a recent study using an avian model explored the effects of MeHg on avian parental behaviour such as nest building, incubation behaviour and provisioning behaviour [68]. Zebra finches were exposed to 0 or 1.2 ppm wet weight MeHg through lifetime dietary exposure. MeHg exposed pairs spent less time constructing nests and built lighter nests, but both variables were also influenced by male age and mass [68]. Control pairs had a greater proportion of successful nest-building trips (pieces of hay brought to the nest compared to number of attempts), but did not differ in amount of hay compared to MeHg-treated finches [68]. As such, the authors suggest a potential compensatory effect of more nest-building trips [68].

#### 3.1.2. Maternal Brain

In the MeHg and retinyl palmitate (Vitamin A) study, hippocampal catalase and glutathione peroxidase activity were also investigated. These two enzymes have critical functions in reducing oxidative damage by detoxication of hydrogen peroxide [69]. Hippocampal catalase activity was reduced in both the MeHg and VitA groups but not the combination treatment. Glutathione peroxidase activity also decreased in MeHg-treated rats, as well as in the combination treatment group [65]. In the prefrontal cortex, total reduced thiol content, a key indicator of redox status, was significantly increased in the MeHg–VitA group [65,70]. No significant redox profile changes were observed in the olfactory bulbs of treated dams compared to control dams [65]. These results collectively suggest a potential adaptive response by which the increase in total thiol content acts to decrease the toxic effects of MeHg. Conversely, decreases in both catalase and glutathione peroxidase activity suggest a toxic effect of MeHg on the maternal hippocampal region. Further evaluation of the transcript and protein levels of catalase and glutathione peroxidase would clarify these findings.

Another study examined the effects of MeHg exposure on glutamatergic homeostasis and oxidative stress in the cerebellum of mice [71]. Exposed dams received drinking water ad libitum with 15mg/L of MeHg from PND1 to PND21. The estimated daily dose of MeHg was 8.25 mg/kg body weight based on the liquid intake per day. No differences in cerebellar glutamate uptake, levels of total sulfhydryl groups, nonprotein sulfhydryl groups, and nonprotein hydroperoxide were observed between control and MeHg-exposed dams [71]. Cerebellar catalase activity showed no difference between groups, while glutathione peroxidase activity significantly decreased in MeHg-exposed dams [71]. Thus, this study suggests a slight neurotoxic effect of MeHg, as decreases in glutathione peroxidase activity limit the brain tissue antioxidant capacity, which is consistent with previous findings [72].

### 3.2. Effects of PCBs on the Maternal Brain or Behaviour

#### 3.2.1. Maternal Behaviour

Two studies investigated the effects of PCB 77 (CASRN 32598-13-3) on maternal behaviour. PCB 77 is reported in a wide range of aquatic and mammalian species, and the high toxicity of PCB 77 is attributed to its coplanar structure [73,74]. One study found that dams administered 4mg/kg PCB 77 (s.c.) daily from GD6 to GD18, spent significantly more time in the nest compared to the control group and more time licking and grooming their pups than the control dams or those having received 2 mg/kg/day of PCB 77 [75]. Although the proportion of time nursing was unaffected by the PCB treatments, there was a statistically significant difference in the proportion of total time nursing in the high-crouch posture specifically, between the exposed and control groups [75]. Both PCB-treated groups showed significantly less high-crouch nursing compared to the control group, with no significant difference between the two PCB doses [75]. 

The second study used a cross-fostering design to explore direct and indirect effects of PCB 77 exposure on maternal behaviour [76]. Dams were exposed to corn oil or 2 mg/kg PCB 77 bodyweight/daily (s.c.) from GD16 to GD18 and pups were cross-fostered or raised by their birth mothers, which resulted in four treatment groups: (1) PCB-exposed dams and their pups or PCB-exposed dams and cross-fostered PCB-exposed pups (data combined), (2) PCB-exposed dams and vehicle (oil)-treated pups, (3) oil-treated dams and PCB-exposed pups, (4) oil-treated dams and their pups or oil-treated dams and cross-fostered oil-exposed pups (data combined). When the data from all PCB groups (1, 2, and 3) were combined, it was shown that dams spent significantly more time on the nest compared to the vehicle-only control group [76]. As well, pup grooming and number of nursing bouts were increased in dams from the PCB treatment groups. There were no differences in amount of time nursing between the PCB groups and oil groups, but PCB groups displayed the high-crouch nursing posture significantly less than the oil-only group [76]. 

Another study investigated maternal PCB exposure using an avian model with exposure from one month prior to pairing, lasting until the hatching of eggs [77]. Adult captive kestrel pairs were administered Aroclors 1248, 1254, and 1260 commercial mixtures consisting of multiple PCB congeners [78]. Birds consumed day-old cockerels injected with the Aroclor mixture, thereby intaking 5-7 μg/g body weight PCBs daily. During the incubation period, 8% of PCB-exposed pairs abandoned their clutches prior to hatching compared to 0% of the control pairs, a difference reported to have a medium effect size [77]. 

Dover et al. (2015) used a mixture of PCB 47 (CASRN 2437-79-8) and PCB 77 (equal parts) at 25 mg/kg wet weight dietary exposure from GD0 to parturition to examine effects on maternal behaviour and underlying molecular mechanisms in rat dams [79]. PCB 47 is a non-coplanar congener which is less toxic but more frequently identified in environmental samples than PCB 77 [73]. Although the authors provided values for food intake, estimated PCB exposure from food intake was not reported. The proportion of time spent in low-crouch nursing posture and high-crouch nursing posture significantly increased compared to control dams on PND4 and PND6, respectively [79]. No effects of treatment were found for the remaining maternal behaviours assessed, including active nursing, pup licking, maternal auto-grooming, time off nest, and resting nursing. Nest building was assessed with results showing that PCB-exposed dams used more nesting strips on GD20 compared to the control group, but the overall quality of nest did not differ significantly between groups [79].

#### 3.2.2. Maternal Brain

From the PCB 47 and PCB 77 exposure study, analysis of the maternal hypothalamus revealed an increased expression of the oxytocin receptor *(OXTR)* gene in the PCB-treated dams that fostered PCB-exposed pups and cross-fostered control pups. This receptor and the oxytocin ligand play a key role in mediating the effects of estrogen on the initiation of maternal behaviour [80,81]. Hypothalamic *Cyp1a1* expression did not differ between groups [79]. The CYP1A1 enzyme belongs to the cytochrome P450 (CYP450) family of enzymes, which metabolize xenobiotic substances and certain endogenous compounds [82,83]. As such, this study suggests that PCB exposure affects *OXTR* expression, which in turn may affect maternal behaviour. 

Another study examined CYP1A1/2 AND CYP1B1/2 protein expression following PCB exposure. Rat dams were administered a mixture of PCB 138 (CASRN 35065-28-2), 153 (CASRN 35065-27-1), 180 (CASRN 35065-29-3), and 126 (CASRN 57465-28-8) from GD15 to GD19 at 0 or 10 mg/kg/day (s.c.) [84]. PCB 138, 153, and 180 are highly abundant noncoplanar congeners, whereas PCB 126 is less abundant but highly toxic [73,85,86,87]. PCB exposure did not induce higher CYP1A or CYP2B expression compared to the control dams, as determined by protein analysis in total brain samples [84]. Similar to the Dover et al., this study reports no changes in CYP450 metabolism of PCBs in the maternal brain.

Honma et al. (2009) used the PCB 153 congener to study alterations in neurotransmitter levels and their metabolites [88]. Dams were administered PCB 153 at 0, 16, or 64 mg/kg body weight by daily oral gavage treatment through GD10 to GD16. Multiple brain regions were analysed including the occipital cortex and hippocampus, which displayed significant decreases in dopamine (DA), DOPAC, and homovanillic acid (HVA) levels for each PCB-treated group compared to the control [88]. In the striatum, HVA levels decreased significantly in the higher dose PCB group. In the hypothalamus, HVA and HVA/DA ratios decreased significantly in the high dose PCB group, while serotonin levels increased significantly in the same group. In the medulla oblongata, DA levels were significantly decreased in the high dose PCB group [88]. Many other neurotransmitter levels and ratios were altered but did not reach statistical significance.

### 3.3. Effects of OCPs on the maternal Brain or Behaviour

#### 3.3.1. Maternal Behaviour

Three studies investigating OCP exposure focused on maternal behaviour outcomes. Matsuura et al. conducted a reproductive toxicity study using lindane, a pesticide which has been banned for agricultural use [89]. Rat dams were given a diet including 0, 10, 60, or 300 ppm lindane for 10 weeks before mating until PND21. Based on daily food intake, the 10 ppm group consumed 0.573 ± 0.0328 and 1.525 ± 0.075 mg lindane/kg body weight per day during the gestational and lactational periods, respectively. The 60 ppm group consumed 3.389 ± 0.167 and 8.941 ± 0.677, while the 300 ppm lindane group consumed 16.55 ± 0.95 and 45.21 ± 3.54 mg/kg body weight per day during the gestational and lactational periods, respectively. Lack of retrieval behaviour and consequential litter loss were observed in one 300 ppm lindane-exposed dam [90]. Other than the one case, lindane exposure did not affect any maternal behaviours, including lactation, nest building, and cannibalism [90]. 

Another study using methoxychlor, a synthetic OCP, examined the effects of maternal exposure from GD11 to GD17 [91]. Doses of methoxychlor at 0, 20, 200, or 2000 μg/kg body weight/day were administered to dams by oral administration from a modified syringe. Compared to the control dams, dams exposed to the lowest methoxychlor dose spent less time nursing, less time in the nest, more time eating and resting outside the nest during the dark period [91]. Within-group post hoc analysis revealed early onset decline in maternal behaviour of the methoxychlor-exposed dams. Compared to PND2, control dams spent less time nursing and in the nest from PND11 onwards while the lowest dose methoxychlor group showed these behavioural changes from PND4 onwards. Similarly, control dams increased time eating and resting at PND15, and the lowest dose methoxychlor group displayed these increases at PND5 for eating and PND7 for resting [91]. 

One study in this scoping review assessed maternal toxicant exposure in humans [92]. Four assessments were used to evaluate maternal psychopathologies including the Brief Symptom Inventory (BSI), Postpartum Bonding Questionnaire (PBQ), Mother to Infant Bonding Scale (MIBS), and Edinburgh Postnatal Depression scale. High scores on these assessments suggest maternal psychopathologies or infant bonding issues. Breast milk was analysed at the eighth month postpartum for 12 OCPs. Of the 12 OCPS, heptachlor epoxide levels positively correlated with PBQ scores, MIBS scores, and three indexes of the BSI, including the global severity index, positive symptom total index, and positive symptom distress index [92]. As well, five subscales of the BSI correlated positively with heptachlor epoxide levels, specifically somatization, depression, anxiety, hostility, and phobic anxiety [92]. Note that this study was included even though our search was not specific to maternal psychopathologies; a search with keywords specific to maternal psychopathologies may have yielded additional results.

#### 3.3.2. Maternal Brain

No data were found pertaining to the effects of OCP exposure on the maternal brain.

### 3.4. Effects of Toxicant Co-Exposure on the Maternal Brain or Behaviour

#### 3.4.1. Maternal Behaviour

One study conducted in Scandinavia exposed mouse dams to an environmentally relevant mixture of 29 organic pollutants, including multiple PCB congeners, OCPs, brominated compounds, and perfluorinated compounds [93]. Dams were exposed to 0, 5000, or 100,000 times the estimated daily intake for humans through dietary feed. Exposure began when the dams were young pups, from weaning through the duration of their pregnancies to project completion. The open field test was used to examine anxiety-like behaviours and locomotion. Exposure to the POP mixture at either dose had no effect on the behavioural endpoints for dams, including time spent within the different zones, total distance moved, or velocity [93]. Note that brominated or perfluorinated compounds may have impacted any effects of the PCBs or OCPs as they were not studied in isolation. 

Another study investigated toxicant co-exposure using Glaucous gull pairs in two different Norwegian breeding regions [94]. Blood concentrations of multiple toxicants including 8 PCB congeners, *p,p*’-DDE, HCB and oxychlordane were measured, and avian parental behaviours were assessed. PCB concentrations in parental pairs were significantly related to the proportion of time away from the nest when not incubating. As well, increased PCB concentrations were related to the number of absences from the nest [94]. These data were later reanalysed and both PCB and oxychlordane blood concentrations were significantly and positively correlated with time away from the nest when not incubating [95]. No significant effects of *p,p*’-DDE or HCB were reported [95]. 

#### 3.4.2. Maternal Brain

Two studies used rodent models to identify potential neurotoxic effects of co-exposure to MeHg and PCB 153 on the cholinergic system. In the first study, rat dams were exposed to 0, 0.5, or 1.0 mg/kg MeHg body weight per day alone or in combination with PCB 153 treatment at 20 mg/kg/day [96]. MeHg treatment spanned GD7 to PND7, while PCB was administered from GD10 to GD16. These dosages and exposure periods followed those used in previous studies demonstrating neurochemical and behavioural changes in adult rats [97,98,99,100]. Dams from the higher exposure MeHg group, the PCB group, and both co-exposed groups each had significant increases in muscarinic receptor (MR) density in the cerebral cortex compared to the control group [96]. In the cerebellum, MR density significantly increased in the high dose MeHg group, while PCB 153 exposure resulted in significantly decreased MR density compared to the control group. Both co-exposure groups had significant decreases in MR density similar to the PCB group [96]. No significant changes in MR density were observed for the low dose MeHg-exposed group in the cerebral cortex or cerebellum. The hippocampal and striatal brain regions did not express any changes in MR density following any treatment. Treatment did not affect the MR dissociation constant in any brain area [96].

Roda et al. [101] used the same dosing regime as the above study to further investigate the potential role of alterations in the cholinergic systems as biomarkers for MeHg and PCB-associated neurotoxicity. In this experiment, dams exposed to the higher dose of MeHg expressed a significant increase in cerebellar MR density [101] (Table 3). Exposure to the lower dose of MeHg, PCB 153, and either MeHg dose in combination with PCB 153 did not result in significant MR density changes. Again, MR dissociation constants did not differ between groups. As well, monoamine oxidase B activity did not differ between the treatment groups and the control group [101].

## 4. Discussion

This scoping review demonstrated the limited number of studies investigating behavioural and neurochemical changes resulting from maternal toxicant exposure. Limited changes in maternal behaviour were reported for MeHg-treated dams, while many behavioural changes were observed in maternal PCB exposure studies. Animal studies investigating OCPs focused on behavioural assessments in which effects were observed at the lowest and highest doses of two different OCPs. One human study showed a positive correlation between OCP levels and maternal psychopathology assessments. Studies involving co-exposure to multiple toxicants were limited in behavioural findings, with the exception of the correlational avian research study. 

Two maternal MeHg exposure studies described change in redox status of the brain, where both increases and decreases were documented. Changes in neurotransmitter levels, gene expression, and protein expression in the brain were reported in three PCB exposure studies. Two co-exposure studies described alterations to the cholinergic system in multiple brain regions, and there were no studies that reported effects of OCPs on neurochemical measures in this review. Note that to our knowledge, no studies have investigated the effects of the mixture considered the Northern Contaminant Mixture on maternal behaviour or related modifications to the brain.

### 4.1. Changes to Maternal Behaviour

#### 4.1.1. Rodent Maternal Care Behaviours

Rat dams exposed to PCB 77 at a higher dose (4mg/kg bw/day) spent more time licking and grooming the pups when compared to dams not exposed to PCBs [75]. Similar findings were shown in a cross-foster design study whereby combined data from PCB-exposed rat dams (2 mg/kg bw/day) rearing PCB-exposed pups and non-exposed dams rearing PCB-exposed pups displayed increases in time grooming as well as in licking and nursing bouts [76]. Recent literature using a glysophate-based herbicide (Roundup) presents similar increases in maternal licking behaviour [102]. Conversely, maternal bisphenol A exposure has shown significant reductions in licking and grooming behaviour [103]. Maternal behaviours including licking and grooming have been shown to be stable across litters, so changes in these behaviours may function to mediate harmful effects of early environmental stressors [57].

The same PCB 77 groups above also spent more time on the nest [75,76]. On the contrary, results from an OCP exposure study showed that mouse dams exposed to low dose methoxychlor spent less time in the nest [91]. Mouse dams treated with bisphenol A have similarly shown increases in time spent out of their nest [104]. Total amounts of maternal care in the methoxychlor study were not different between treatment groups, but alterations in the onset and decline of maternal behaviours were observed.

PCB groups in both studies showed significantly less high-crouch nursing [75,76]. Contradicting results were shown with a dietary exposure study using PCB 47 and 77, as a proportion of time spent in high-crouch nursing posture increased on PND6 [79]. As the authors noted, these results could be attributable to procedural differences and different mechanisms of toxicity of the two PCB congeners. 

#### 4.1.2. Rodent Maternal Exploratory Behaviours

One study found effects of MeHg on exploratory behaviour, as the highest MeHgCl dietary dose group showed reduced exploratory behaviour compared to MeHgCys and the vehicle-treated control [66]. MeHgCl is commercially available and commonly used in neurotoxicology studies, but MeHgCys has been shown as the dominant chemical form in fish tissue, to which humans are exposed through consumption [105]. As MeHgCl is more hydrophobic than other MeHg forms, this may cause differential toxic properties and may limit environmental relevance [105]. From studies included in this scoping review, all but two studies used MeHgCl in their treatment protocol [66,68]. 

#### 4.1.3. Avian Parental Behaviours

Treatment with MeHg was explored in an avian model using zebra finches. Findings showed alterations in nest-building behaviour, a key component of avian paternal behaviour [106]. As well, a compensatory effect was suggested because MeHg-exposed birds had fewer successful nest-building trips but did not differ in the amount of hay brought to the nest building area, per hour. Behavioural data in this study were analysed in reproductive pairs, as zebra finches exhibit a biparental care strategy [107]. Interspecies comparison is limited due to different parental strategies, but these behavioural changes should be noted.

Another avian study found that PCB-exposed American kestrels abandoned their clutch more than control parents [77]. Incubation behaviour is a critical component of offspring success in this species as optimal development of an embryo occurs within a small temperature range [108,109]. Incubation behaviour has been associated with increases in serum prolactin level [110], which may contribute to the molecular pathways underlying these behavioural differences [111]. Similarly, disruptions to endocrine signalling may contribute to the correlations observed between toxicant blood concentrations of Glaucous gulls and non-incubating time away from the nest, as suggested by the authors [94,95]. As well, they note neurological disruptions as a potential cause of these behavioural changes [94,95]. 

#### 4.1.4. Human Maternal Psychopathologies

In human mothers, heptachlor blood concentrations were associated with maternal psychopathological assessments, including correlations with depression and anxiety measures. Maternal anxiety and depression have been associated with reduced infant care, so heptachlor exposure may have negative effects on maternal care [112,113].

### 4.2. Neurochemical Changes in the Maternal Brain

#### 4.2.1. Redox Activity

Changes in hippocampal catalase and glutathione peroxidase were observed in one MeHg exposure study, along with changes in thiol content in the prefrontal cortex [65]. Similarly, another study showed reduced cerebellar glutathione peroxidase activity following MeHg exposure [71]. The cellular mechanisms underlying MeHg neurotoxicity are not fully understood, but evidence suggests the electrophilic properties of MeHg allow for interactions with key nucleophilic groups including thiols and selenols. Furthermore, these interactions may disrupt the activity of many important metabolic proteins and receptors involved in antioxidant defense mechanisms [114].

#### 4.2.2. OXTR Gene Expression

Oxytocin modulates the onset of maternal behaviour, which is mediated by receptor binding [115,116]. Dover et al. reported elevation of *OXTR* expression in PCB-exposed mice. This is consistent with the behavioural findings where high-crouch licking posture was increased on PND6. High-crouch posture is associated with optimal milk letdown, a key component of maternal care [117].

#### 4.2.3. P450 Protein Expression

One study observed no change in hypothalamic *Cyp1a1* expression following PCB 47 and 77 treatment [79]. Cyp1a1/2 and Cyp1b1/2 protein analysis was conducted in another PCB mixture study and expression levels were comparable between control and treated dams [84]; however, caution must be expressed with respect to the implication of these results given that the route of administration was a subcutaneous one and thus, unlikely a natural route of exposure to these substances. 

#### 4.2.4. Muscarinic Receptor Density

Maternal rat dams treated with higher doses of MeHg showed increase in MR density in the cerebral cortex and cerebellum, while PCB-exposed dams showed an increase in cerebral cortex MR density but a decrease in the cerebellum [96]. Co-exposed groups had similar regional increases and decreases as the PCB-only exposed group. In a second study using the same dosing regimen, a higher dose of MeHg increased cerebellar MR density, but limited changes were observed with the other treatment groups [101]. 

#### 4.2.5. Neurotransmitter Levels

Decreases in dopamine and its metabolites were observed in multiple brain regions [88] following PCB exposure. In the same study, serotonin levels significantly increased in the hypothalamus. Both dopamine and serotonin are involved in molecular pathways underlying maternal behaviour, [118,119] so changes in levels of these neurotransmitters may have consequential effects on maternal care. 

### 4.3. Limitations and Implications

#### 4.3.1. Toxicant Inclusion

This scoping review focused on exposure to MeHg, PCBs, and OCPs, as these toxicants are the most abundant in the Northern Arctic region. As such, keywords were used to identify studies that assessed exposure to these three chemicals classes. The specificity of these keywords may have missed relevant studies, such as avian studies measuring mercury levels, which are a reliable proxy for MeHg levels [120]. 

#### 4.3.2. Differentiating Direct and Indirect Effects of Exposure

Maternal behaviour can be affected by direct exposure to environmental toxicants [75,91]. Pup behaviour can also have direct effects on maternal responsiveness and behaviour [121]. Results from this scoping review show multiple neurochemical changes following toxicant exposure. As well, one cross-fostering designed study found significant increases in maternal behaviours when data from PCB-exposed dams raising PCB-exposed pups or control pups, and control dams raising PCB-exposed pups were combined [76]. These data suggest both direct effects on maternal behaviour and pup-mediated effects, which have been shown in other toxicant exposure studies [122]. Research protocols should continue to employ cross-fostering designs to gain insight into direct and pup-interaction behavioural changes.

#### 4.3.3. Environmental Relevance

Humans are exposed to multiple toxicants through dietary and environmental sources [30]. Only six of 16 experimental studies assessed toxicant co-exposure. While it is crucial to understand molecular changes underlying single toxicant exposure, these studies lack insight into potential synergistic, additive, or antagonistic effects of environmentally relevant co-exposure. Future studies should aim to include co-exposure groups for highly abundant toxicants. 

## 5. Conclusions

This scoping review gives insight into the multitude of effects associated with maternal toxicant exposure. Varied behavioural effects were identified following MeHg, PCB, or OCP treatment in avian and rodent models. The neurochemical pathways so far shown to be involved in mediating the effects of toxicant exposure include the oxytocin, serotonin, and dopamine signalling pathways, antioxidants, and muscarinic receptors. Future environmental toxicant research needs to characterize the potential harmful or adaptive responses to toxicant exposure involving neuromodulator signalling pathways and the role of antioxidants in these. Detailed mechanistic studies investigating these pathways and maternal behavioural endpoints are necessary, given the critical intersection between maternal behaviour and offspring development, with an outlook toward sequential pregnancy and trans-generational consequences. Findings from this scoping review may help guide research and inform policy decisions in this field. 

## Figures and Tables

**Figure 1 ijerph-17-00926-f001:**
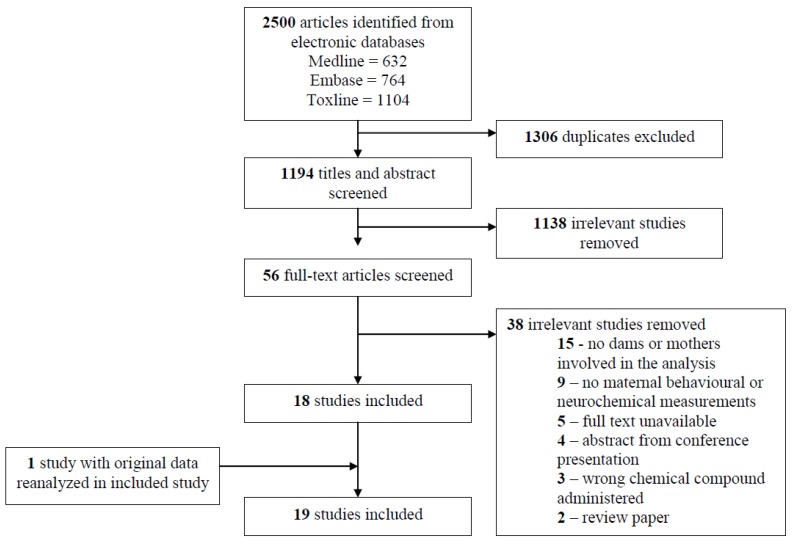
Flow diagram for the scoping review study selection process.

**Table 1 ijerph-17-00926-t001:** Keywords and subject headings used for Medline and Embase searches.

	Keywords	Subject Headings (MeSH)	Subject Headings (Emtree)
Concept 1: Population	pregnancy.ti,ab,kw. pregnant.ti,ab,kw. perinatal.ti,ab,kw. maternal.ti,ab,kw. postpartum.ti,ab,kw. antenatal.ti,ab,kw. gestation*.ti,ab,kw.	pregnancy OR perinatal care OR maternal exposure OR postpartum period	pregnancy OR perinatal period OR perinatal care OR maternal care OR maternal exposure
Concept 2: Exposure	methylmercury.ti,ab,kw. MeHg.ti,ab,kw. polychlorinated biphenyl*.ti,ab,kw. PCB*.ti,ab,kw. Aroclor*.ti,ab,kw. kanechlor*.ti,ab,kw. clophen*.ti,ab,kw. phenoclor*.ti,ab,kw. pyralene*.ti,ab,kw. fenclor*.ti,ab,kw. sovol*.ti,ab,kw. chlorfen*.ti,ab,kw. delor*.ti,ab,kw. organochlorine pesticide*.ti,ab,kw. aldrin*.ti,ab,kw. chlordan*.ti,ab,kw. chlordecone*.ti,ab,kw. chloroacetate*.ti,ab,kw. chlorobenzene*.ti,ab,kw. chlorofluorocarbon*.ti,ab,kw. chloroform*.ti,ab,kw. chloromethane*.ti,ab,kw. DDT*.ti,ab,kw. dichlorodiphenyldichloroethylene*.ti,ab,kw. dichlorodiphenyldichloroethane*.ti,ab,kw. dieldrin*.ti,ab,kw. endrin*.ti,ab,kw. ethyl chloride*.ti,ab,kw. ethylene dichloride*.ti,ab,kw. heptachlor*.ti,ab,kw. hexachlorocyclohexane*.ti,ab,kw. methoxychlor*.ti,ab,kw. methylchloride*.ti,ab,kw. methylene chloride*.ti,ab,kw. mirex*.ti,ab,kw. tetrachloroethylene*.ti,ab,kw. toxaphene*.ti,ab,kw. Trichloroepoxypropane*.ti,ab,kw. Trichloroethane*.ti,ab,kw. Trichloroethylene*.ti,ab,kw. northern contaminant mixture*.ti,ab,kw. NCM*.ti,ab,kw. (north* adj3 (pollutant* or toxicant* or contaminant*)).ti,ab,kw.	methylmercury compounds OR polychlorinated biphenyls OR Aroclors OR hydrocarbons, chlorinated OR aldrin OR chlordane OR chlordecone OR chloroform OR ddt OR dichlorodiphenyl dichloroethylene OR dichlorodiphenyldichloroethane OR dichloroethylenes OR dieldrin OR endrin OR ethyl chloride OR ethylene dichlorides OR heptachlor OR hexachlorocyclohexane OR methoxychlor OR methyl chloride OR methylene chloride OR mirex OR tetrachloroethylene OR toxaphene OR trichloroepoxypropane OR trichloroethanes OR trichloroethylene	methylmercury OR polychlorinated biphenyl OR Aroclor OR Aroclor 1242 OR Aroclor 1260 OR Aroclor 1254 OR Aroclor 1248 OR organochlorine insecticide OR “1,1 dichloro 2,2 bis(4 chlorophenyl)ethane” OR “1,1 dichloro 2,2 bis(4 chlorophenyl)ethylene” OR “1,1,1 trichloro 2 (2 chlorophenyl) 2 (4 chlorophenyl)ethane” OR 1,2 dichlorobenzene OR 1,4 dichlorobenzene OR aldrin OR alpha hexachlorocyclohexane OR beta hexachlorocyclohexane OR campheclor OR chlordane OR chlordecone OR chlorphenotane OR clofentezine OR dieldrin OR endosulfan OR endrin OR heptachlor OR heptachlor epoxide OR isobenzan OR lindane OR methoxychlor OR mirex OR nonachlor OR oxychlordane OR photomirex OR organochlorine pesticide OR chlornitrofen OR chlorobenzilate OR chloropicrin OR chlorothalonil OR chlorthiamid OR dacthal OR dicofol OR tetradifon OR chlorinated hydrocarbon
Concept 3: Outcome	brain*.ti,ab,kw. behavio?r*.ti,ab,kw.	behavior OR maternal behavior OR brain	behavior OR maternal behavior OR brain

**Table 2 ijerph-17-00926-t002:** Keywords used for Toxline searches.

	Keywords
Concept 1: Population	pregnancy OR pregnant OR perinatal OR maternal OR postpartum OR antenatal OR gestation OR gestational
Concept 2: Exposure	methylmercury OR MeHg OR polychlorinated biphenyl OR pcb OR Aroclor OR kanechlor OR clophen OR phenoclor OR pyralene OR fenclor OR sovol OR chlorfen OR delor OR organochlorine pesticide OR aldrin OR chlordan OR chlordecone OR chloroacetate OR chlorobenzene OR chlorofluorocarbon OR chloroform OR chloromethane OR ddt OR dichlorodiphenyldichloroethylene OR dichlorodiphenyldichloroethane OR dichloroethylene OR dieldrin OR endrin OR ethyl chloride OR ethylene dichloride OR heptachlor OR hexachlorocyclohexane OR methoxychlor OR methylchloride OR methylene chloride OR mirex OR tetrachloroethylene OR toxaphene OR trichloroepoxypropane OR trichloroethane OR trichloroethylene OR northern contaminant mixture OR ncm OR northern pollutant OR northern toxicant OR northern contaminant
Concept 3: Outcome	behavior OR behaviour OR brain

**Table 3 ijerph-17-00926-t003:** Details of studies included in this scoping review.

Reference Number	Study	Toxicant(s) of Interest	Objective	Study Design	Maternal Subjects	Toxicant Treatment Group(s)	Exposure Route	Exposure Period	Behavioural Findings	Neurochemical Findings
[65]	Espitia-Pérez et al., 2018	MeHg	Examine the effects of MeHg and VitA co-exposure on pregnant and lactating Wistar rats to evaluate behavioural and biochemical changes in brains of the dams and their offspring	Experimental design	Wistar rats N = 30	0.5mg/kg body weight/day MeHg	Oral gavage treatment	GD0 to PND21	No differences in nursing and pup retrieval behaviours were observed between treated dams and controls.	Hippocampal catalase activity was reduced in both the MeHg and VitA groups, while glutathione peroxidase activity decreased in both the MeHg and MeHg-VitA-treated groups. In the prefrontal cortex, total reduced thiol content significantly increased in the MeHg-VitA group. No significant redox profile changes were observed in the olfactory bulbs of treated dams.
[93]	Hudecova et al., 2018 ^1^	PCBs: 28, 52, 101, 118, 138, 153, 180 OCPs: *p,p’*-DDE, HCB, α-chlordane, oxychlordane, *trans*-nonachlor, α-HCH, β-HCH, ϒ-HCH, dieldrin 7 BFRs6 PFAAs	Determine whether a POP mixture relevant to human exposure levels affects basal corticosterone levels, anxiety-like behavior, and locomotor activity in female mice and their offspring	Experimental design	129:C57BL/6F0 hybrid female mice N = 47	5000 or 100,000x EDI toxicant mixture	Dietary exposure	From weaning prior to mating until project completion	Exposure to the POP mixture at either dose had no effect on the endpoints of the open field behavioural test for dams including: time spent within the different zones, total distance moved, or velocity.	NR
[68]	Chin et al., 2017	MeHg	Determine the effects of MeHg exposure on avian parental behavior and reproductive success in zebra finches	Experimental design	Zebra finch pairs N=87 (N = 73 initiated nests)	1.2ppm MeHg wet weight	Dietary exposure	From in ovo through maturity	MeHg-exposed pairs spent less time constructing and built lighter nests (both influenced by male age and mass). Control pairs had greater efficiency in bringing hay to the nest, but did not differ in amount of hay compared to MeHg-treated finches, suggesting a compensatory effect of more trips made by the MeHg-treated finches.	NR
[79]	Dover et al., 2015 ^2^	PCBs: 47 and 77	Examine possible molecular mechanisms underlying changes in maternal care behaviour due to PCB exposure	Experimental design	Sprague–Dawley rats N=11	25mg/kg wet weight PCB 47 and 77	Dietary exposure	GD0 to PND0	PCB altered nest building and maternal care behaviours. Specifically, there was a significant increase in time spent in low crouch and high crouch nursing posture on PND4 and PND6 respectively.	Molecular analysis revealed an increased *OXTR* expression in the hypothalamus of dams exposed to PCBs.
[92]	Yalçin et al., 2015	OCPs: α-HCH, β-HCH, ϒ-HCH, aldrin, dieldrin, heptachlor, heptachlor epoxide, α-endosulfan, β-endosulfan, trans-chlordane, cis-chlordane, DDT	Assess detectable OCPs in maternal breast milk to evaluate the relation between OCPs and maternal psychopathologies	Correlational design	Human mothers N=75	NR	NR	NR	Heptachlor epoxide levels were positively associated with PBQ scores, MIBS scores, and three indexes of the maternal BSI (the global severity index, positive symptom total index and positive symptom distress index) and five subscales of the maternal BSI (depression, hostility, anxiety, phobia, and somatic symptoms).	NR
[64]	Weston et al., 2014	MeHg	Examine the effects of MeHg and prenatal stress on maternal and infant behaviour and neurochemical markers	Experimental design	Long–Evans rats N = 66 (N = 24 for behavioural testing)	0.5 or 2.5 ppm MeHg	Drinking water	2 to 3 weeks before breeding to post-weaning	Changes in maternal behavior, attributed to MeHg exposure alone, were extremely limited.	NR
[101]	Roda et al., 2012	MeHg PCB 153	Evaluate brain and lymphocyte muscarinic receptors and cerebral monoamine oxidase-B activity as potential biomarkers for assessing exposure to environmental toxicants	Experimental design	Sprague–Dawley rats N = 12 per set of experiment	0.5 or 1mg/kg body weight/day MeHg and/or 20mg/kg/day PCB 153	Drinking water (MeHg) Oral gavage treatment (PCB 153)	GD7 to PND7 (MeHg) GD10 to GD16 (PCB 153)	NR	Cerebellar muscarinic receptor density increased (87%) with exposure to the higher MeHg dose, while no changes were observed in the lower MeHg dose, PCB dose or PCB coexposed group. The muscarinic receptor (MR) cerebellar dissociation constants were not altered in any of the treatment groups. Cerebellar MAO-B activity did not differ between any treatment group and the control.
[84]	Bonfanti et al., 2009 ^3^	PCBs: 138, 153, 180 and 126	Investigate PCB disposition in two maternal and fetal rat organs for toxic implications	Experimental design	Sprague–Dawley rats N=10	10mg/kg body weight/day PCB 138, 153, 180, 126 mixture	Subcutaneous injection	GD15 to GD19	NR	CYP1A and CYP2B levels were determined in maternal brains.
[66]	Glover et al., 2009	MeHg	Compare the effects of MeHgCl and MeHgCys on the accumulation, brain gene expression, and behavior of mice	Experimental design	Balb/c mice N=32 (N= 31 due to MeHgCys adverse toxicity)	1.5 or 4.5 mg/kg MeHgCys or MeHgCl	Dietary exposure	6 weeks prior to mating through to 2 weeks following birth	High MeHgCl diet group exhibited reduced exploratory behaviour compared to the control, and increased latency to move compared to the control and high MeHgCys exposure groups.	NR
[88]	Honma et al., 2009	PCB 153	Investigate the effects of PCB administration on cerebral neurotransmitters and related substances in rat dams and offspring	Experimental design	Crj:CD(SD)IGS rats N=30	16 or 64 mg/kg/body weight/day PCB 153	Oral gavage treatment	GD10 to G16	NR	In the occipital cortex, DA, DOPAC, and HVA levels decreased in both PCB-treated groups. In the hippocampus, DA, DOPAC, and HVA levels decreased by 30% and 40% in the 16 mg/kg/bw and 64 mg/kg/bw, respectively. In the striatum, HVA decreased significantly in the 64 mg/kg/bw group. In the hypothalamus, HVA and HVA/DA ratios decreased significantly in the 64 mg/kg/bw day group. 5HT increased significantly in the same group. In the medulla oblangata, DA level decreased significantly in the 64 mg/kg/bw group.
[96]	Coccini et al., 2006	MeHg PCB 153	Determine whether MeHg and PCB 153 alter the MRs in the cerebral cortex, cerebellum, hippocampus and striatum	Experimental design	Sprague–Dawley ratsN=24 per set of experiment	0.5 or 1.0mg/kg/day MeHg and/or 20mg/kg/day PCB 153	Drinking water (MeHg) Oral gavage (PCB 153)	GD7 to PND7 (MeHg) GD10 to GD16 (PCB 153)	NR	Cerebral cortex MR density increased for 1.0 mg/kg/day MeHg, PCB 153, and both MeHg+PCB153 treatments groups (60% MeHg group, 47% PCB group, 45% 1.0 MeHg/kg/day+PCB153, 42% 0.5MeHg/kg/day+PCB153). Treatment with MeHg at the higher dose resulted in increased cerebellar MR density (87%), while PCB 153 exposure resulted in significantly decreased MR density (27%). Both combined exposure groups resulted in MR density similar to the PCB-exposed group. The lower MeHg dose did not cause any changes in MR density in the cerebellum of dams. In the hippocampus and striatum, no MR density changes were observed following any treatment or combination of treatments. In all brain areas, the dissociation constant values for MR were not altered.
[95]	Bustnes et al., 2005	PCBs: 99, 118, 138, 153, 170, and 180 3 OCPs: Oxychlordane, *p,p*’-DDE, HCB	Analyse four fitness components (time spent away from nest, early chick growth and return rate) in relation to blood residues of PCBs, OCPs in Glaucous gulls	Correlational design	Glaucous gulls N=16	NR	NR	Lifetime	PCBs and oxychlordane were positively and significantly related to time spent away from the nest site when not incubating. DDE and HCB levels had no effect on this trait.	NR
[62]	Cummings et al., 2005	PCB 77	Differentiate between direct and indirect effects of PCB exposure on maternal behaviour	Experimental design	Long-Evans rats N=36	2mg/kg body weight/day PCB77	Subcutaneous injection	GD6 to GD18	Dams exposed to PCBs during pregnancy spend more time on the nest and more time grooming (and licking) the pups, when compared to control dams.	NR
[90]	Matsuura et al., 2005	OCP: ϒ-HCH (lindane)	Assess the endocrine disruption activity and toxicity of lindane using additional toxicological and behavioral endpoints	Experimental design	Crj:CD(SD)IGS female ratsN=24	10, 60, or 300 ppm lindane	Dietary exposure	10 weeks before mating until PND21	Lindane exposure did not affect any maternal behaviours including lactation, nest building and cannibalism. Lack of retrieval behaviour and consequential litter loss was observed in one 300ppm lindane-exposed dam.	NR
[75]	Simmons et al., 2005	PCB 77	Investigate the effects of PCB exposure on the behavior of dams as they rear exposed litters	Experimental design	Long–Evans rats N=21 (N=19 due to failed delivery of 2 dams)	2mg/kg body weight/day or 4mg/kg/body weight/day PCB 77	Subcutaneous injection	GD6 to GD18	PCB 77 exposure resulted in changes in maternal behaviour including: an increase in time spent on the nest, increase in licking and grooming of the offspring, and a decrease in the display of the high-crouch nursing posture.	NR
[71]	Manfroi et al., 2004	MeHg	Investigate the effects of lactational MeHg exposure on neurotoxicity and glutamatergic transmission	Experimental design	Swiss Albino mice N=14	15mg/l MeHg	Drinking water	PND1 to PND21	NR	MeHg exposure did not alter glutamate uptake in cerebellar slices of dams. Cerebellar levels of total and nonprotein sulfhydryl groups and nonprotein hydroperoxide did not differ between control and MeHg-treated dams. MeHg exposure inhibited the activity of cerebellar glutathione peroxidase but had no effects of cerebellar catalase activity.
[77]	Fernie et al., 2003 ^4^	PCBs: Aroclors 1248, 1254, 1260	Identify short and long-term abnormal development and behavior of American kestrels through all stages of the breeding season from parental PCB exposure	Experimental design	American kestrel pairs N= 50	5-7μg/g body weight/day Aroclor 1248, 1254, 1260 mixture	Dietary exposure	1 month prior to pairing until anticipated hatching of eggs	8% of PCB-exposed pairs abandoned their clutches prior to hatching. There were no incidences of altered incubation behavior in the PCB-exposed pairs of the next breeding season.	NR
[91]	Palanza et al., 2002	OCP: methoxychlor	Investigate the effects of maternal exposure to methoxychlor on behaviour responses of dams and their offspring	Experimental design	CD-1 mice N=72-84 treated (N=64-80 for data analysis)	20, 200, or 2000 μg/kg body weight/day methoxychlor	Modified oral gavage treatment	GD11 to GD17	Dams exposed to the low dose of methoxychlor spent lower amounts of time in the nest, nursing and more time eating and resting outside the nest during the dark period.	NR
[94]	Bustnes et al., 2001	PCBs: 28, 101, 99, 118, 138, 153, 170 and 180	Investigate the effects of PCB contamination on nesting behaviour in Glaucous gulls	Correlational design	Glaucous gulls N=16	NR	NR	Lifetime	Time away from the nest and proportion of time absent from the nest was significantly related to PCB concentration in blood.	NR

NR: not reported; Note: subgroups of the maternal subjects were used in multiple studies as indicated by the sample sizes in brackets. 1: Relative amounts of the 29 toxicants are based on the estimated daily intake of Scandinavians. 2: Equal parts PCB 47 and 77 were added to the chow diet. 3: Equal concentration of PCB 138, 153, and 180 were included in the PCB mixture, and PCB 126 was added at a 1:10000 ratio. 4: Equal weight of Aroclors 1248, 1254, and 1260 were added to frozen cockerel diet.

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
