# Peer review of "Canadian Arctic Contaminants and Their Effects on the Maternal Brain and Behaviour: A Scoping Review of the Animal Literature"

_ijerph, 2020, doi:10.3390/ijerph17030926_

Round 1

Reviewer 1 Report

The authors have addressed this reviewer's comments in the revised version and have improved the conclusions from the studies reviewed. I just have these minor edit recommendation

1. Abstract, line 28. It states ...",although there remains gaps in the literature."

I would suggest that one or two of these gaps be identified in this sentence.

2. Section 1.3 Organochlorine pesticides, lines 65-66, it states "OCPs persist in the environment due to their slow metabolism and clearance."

The OCPs are persistent in the environment but it is not because of their slow metabolism and clearance. They are persistent in the environment because of the stability and slow biodegradation not because of their slow metabolism and clearance. Those are the reasons it persists in the body. I recommend changing either the first or second part of the sentence depending on the authors focus.

3. Section 1.5 Maternal brain and behaviour line 112; it states; this paper aims to provide a scoping review of the literature to gain a deeper understanding of the effects of maternal exposure to....."

"gain a deeper understanding" seems strong given the data, not crucial but would suggest softer wording such as "a scoping review of the literature in an attempt to better understand the effects...."

Page 23 line 184. The sentence is an indented paragraph and starts with a capital "No difference...."

This seems to be a continuation of the sentence on page 8 line 177, prior to the table. If so then the indent and capital are not required.

Reviewer 2 Report

This second version of the paper is well written and more clear than the first one. The topic is very interesting: contaminant mixtures uses and their combined negative effects on animal maternal brain and animal maternal behavior are considered. A very big bibliographic research work was done. 

Each paragraph is extended, and more important information are added in it. Now it is clear and many papers were collected. So the paper can be useful as the starting point for other studies.

Reviewer 3 Report

I have gone through this paper entitled “Canadian Arctic Contaminants and Their Effects on the Maternal Brain and Behaviour: A Scoping Review of the Animal Literature” submitted to IJERPH. This review is prepared well. I feel that overall the review has been organized and structured well. I have also gone through the reviewers comments and author’s response. The authors addressed very well according to the reviewer’s comments/suggestions.

I have no further suggestion after seeing/verifying the author's response.  This can be accepted now.

Author Response

This manuscript is a resubmission of an earlier submission. The following is a list of the peer review reports and author responses from that submission.

Round 1

Reviewer 1 Report

Several major and minor deficiencies in this review, as written, are described in the attached comments.

General Comments:

The authors have prepared a scoping review of the effects of exposures to three types of persistent organic pollutants (POPs), namely methylmercury (MeHg), polychlorinated biphenyls (PCBs) and organochlorine pesticides (OCPs) common in contaminated environments on the maternal brain and behaviours, particularly during development. This is a significant problem, suitable for both scoping and systematic reviews.

That said, there are several major deficiencies in the submitted review including the following: i) Abbreviations are not used consistently throughout the review after their introduction (for example, methylmercury and polychlorinated biphenyls appears at line 140 of the review instead of MeHg and PCBs); ii) It is more logical to this reviewer to organize the various articles in Table 3 by year, with the most recent listed first (2019), than alphabetically; iii) the format of references in Table 3 is unsatisfactory. Thus, the references should be listed by number first, and not by author(s) first (see below for more details). It is unnecessarily difficult to find individual references in Table 3 in the bibliography, which is numerical, not alphabetical; and iv) certain of the references are incomplete lacking internet coordinates (see below for more information). Additional issues in the current version of the manuscript are documented below.

Specific Comments:

Line 24:  Abstract: --- indicate that exposures to ---

Line 29: Abstract: --- ecologically relevant toxicants should be replaced with persistent organic pollutants. (POPs) ---

Line 37: Environmental pollution is a global problem ---

Line 39-40: --- polychlorinated biphenyls (PCBs) and organochlorine pesticides (OCPs), two classes of industrial chemicals that persist in the environment ---

Line 49: --- predators at the top of the food chain web. The food supply is much to complex to be described as a chain.

Line 51-52: --- fish consumption as a part of a healthy diet, the chronic low-level dietary intake of MeHg has become more prevalent and as such may pose a significant toxicological problem risk.

Line 61: --- chain web ---

Line 67-68: OCPs are highly persistent in the environment due to their slow metabolism and clearance ---. Chemical structures are not stable; chemicals can be. “structural stability” should be replaced with “due to their slow metabolism and clearance”.

Line 71: --- bioaccumulation and ability potential to cause adverse effects in humans.

Line 83-85: --- higher in populations who consume large amounts of traditional foods from the marine environment than in those who do not.

Line 86: What is an elevated “level” of MeHg? Elevated compared to what? What concentration of MeHg in human hair (a major storage site in humans) is considered to be elevated?

Line 89: When was the Northern Contaminant Program developed?

Line 94: “The NCM was formulated to comprise the 27 most abundant environmental contaminants: 14 PCB 93 congeners, 12 OCPs and MeHg present in the same concentrations found in the blood profiles of 94 mothers residing in the Canadian Arctic.” How many mothers were analyzed to determine these average concentrations of the various POPs found in the blood that are used within the NCM?

Line 140: OCPs appears to be missing from this line.

Table 3, P.7: It would seem more logical to this reviewer to organize the various articles in this table by year, with the most recent listed first (2019) than alphabetically. The most sophisticated articles with relevance to the chemical assays are more likely those that are more recent than not.

Table 3, Bonfanti et al, 2009. A total of 10 mg/kg BW/day of PCBs 138, 153, 180 and 126 in a single mixture was administered SC. What are the relative amounts of the 4 individual PCB congeners? Are they equimolar; equal weight; or other? This should be explained in a subnote to the table. The same is true for the studies of Dover et al, 2015; Fernie et al, 2003; Hudecova et al, 2018.

Line 160: (Results). Rather than adding novel comments about the manuscripts analyzed the authors routinely repeat the conclusion(s) of the researchers who wrote these papers. Such commentary is a characteristic of any comprehensive review article and the authors should make significant changes in this component of their manuscript prior to its reconsideration by the IJERPH. The specific questions below must be considered for change in this regard as well as some general comments.

Thus, the authors make no attempt to distinguish between changes that are neurotoxic vs those that are adaptive (biological responses that result in a decrease in toxicity/pathological response). 

A specific example occurs at lines 200 to line 207. In this study, two enzymes involved in the detoxication (not detoxification, a term which is correctly reserved for alcoholics and drug addicts who are denied access to their favorite poison) of reactive oxygen species are decreased in activity. A state-of-the-art study would have investigated these alterations in catalase and glutathione peroxidase enzyme activity at the protein and mRNA levels to understand mechanistically what is happening i.e. (toxication vs adaptation changes). The authors go on to point out that reduced thiol content of the prefrontal cortex increases under these treatment conditions. Not stated is whether this change in number (increase) of thiol groups occurs in the total thiol pool, the protein thiol pool or the soluble thiol pool. This is a deficiency of the study if such details are absent from the original paper.

The authors of this review are left with the dichotomy where an increase in number of thiol groups is probably an adaptive detoxication response whereas decreases in (specific and/or total) activity of catalase and glutathione peroxidase are likely to toxication (inactivation due to oxidation at the active site of the enzyme) responses.

Line 187: How much food was consumed in this study? Knowing the amount of food consumed is the only way to calculate the dose consumed, a very important component of a well-designed toxicological study .This deserves a comment.

Line 209: Same comment. What was the dose consumed, determined from the amount of MeHg-water consumed?

Line 218: --- the high toxicity of PCB 77 is attributed to its coplanar structure is preferable and more accurate than “its coplanar structure allows for its high toxicity”.

Line 241: Aroclor 1248, etc are correctly capitalized here but not in Table 1 of the manuscript.

Line 249: --- nursing posture significantly increased ---

Line 281: --- using lindane, a toxic pesticide. Toxic is redundant; all pesticides are toxic.

Line 360:--- change in redox status of the brain, where both increases decreases and increases were documented.

Line 487: The References section needs much attention. The reader deserves to have links that take all referenced materials to the original journal and/or to a current internet link. For example, references 11, 12, 14, 15, 18, 22, 28, 33  are incomplete. Moreover, the references in Table 3, P.7 should be listed by number first rather than alphabetically. Thus, Bonfanti et al, 2009 should be shown in the table as 79. Bonfanti et al, 2009. It makes it a terrible waste of time to find references that appear as authors in the table but numerically in the References.

Reviewer 2 Report

This paper is a review of the literature to find information of the effects or 3 environmental toxicants, methylmercury, polycholorinated biphenyls and organochlorine pesticides on maternal behavior. The research question is good and the environmental toxicants investigated are relevant. Unfortunately a review of the available animal studies did not show a clear relationship between maternal behavior and the environmental toxicants. I better relations was seen with biochemical changes in the brain after exposure to these substances. The paper also concludes that little information is available on their research question especially with human data.

General comments

The content of the paper seems to show that environmental contaminants have an effect on the biochemistry of the brain but the behavioural effects are not consistent at least with the information or knowledge we currently have.

I recommend that the authors change their conclusion and try to find a more specific subject area that requires further investigation based on their review as opposed to concluding that there is limited knowledge regarding effects of these ecological toxicants on the maternal brain because the current animal data does not suggest a strong correlation with behaviour. Either that, or that the relationship is very complex. Another option would be to look at the studies more closely and see if there is a reason why some studies show effects an others do not. Maybe it is related to the dose administered, to the specific endpoints investigated or the sensitivity of the species used in the experiments

As with all toxicants, the dose administered or concentration in the body will be key to the effects observed and it seems to me that the doses required to alter maternal behaviour are more likely to have a significant effect on fetal development therefore your research question is looking at a much less sensitive endpoint from exposure to these substances. If this is not the case, then additional information should be added to justify looking at this endpoint.

Section 1.2 polychlorinated biphenyls; This section would benefit from adding text which relates this class of substances to effects on the nervous system.

Section 1.3 Organochlorine pesticides; This section would also benefit from the addition of text which related this class of substance to effects on the central nervous system associated with its effects on acetylcholinesterase.

Section 1.5, Maternal brain and behaviour, lines 103-106. This statement talks about exposure to toxic chemicals affecting maternal behaviour but it is not yet established that this is the occurring.

Section 2.1 Research question. While this question can be studies in animal models, it seems difficult to study in humans. In humans, other factors such as education, socioeconomic factors, mental health, personal preferences and others will all affect maternal behaviour to a greater extent than the presence of environmental contaminants.

Table 3, page 158. I am unsure of the value of adding the study by Bonfanti et al in this table as it was performed with a subcutaneous injection which is not a likely route of exposure for these substances.

Section 3.2.1 Maternal behaviour; When referring to PCB 47 or PCB 77, more information should be provided about the substance, either a chemical name or a CASRN.

Section 3.4.1 Maternal Behaviour; One study is summarized that includes Brominated flame retardants and perfluorinated compounds which were not previously discussed or included in other studies so it is difficult to determine if the effects are the result of the chemicals of interest or other in the mixture.

Section 4.2.4 Human maternal psychopathologies; This paragraph brings in new issues with the psychopathologies and brings in the issue of whether this is the mechanism by which environmental toxins affect maternal care.

Reviewer 3 Report

Title: Effects of Environmental Toxicant Exposure on the Maternal Brain and Behaviour: A Scoping Review.

This paper deals with a very interesting topic, because it takes into consideration an important problem like contaminant mixtures uses and their combined negative effects on maternal brain and maternal behavior. A very big bibliographic research work was done. 

But, reading the abstract it is not possible to understand what about brain and behavior is speaking until “Results” where the reader can understand that only animal models were investigated. I think that also the title should be modified adding the adjective "animal".  

The introduction provides sufficient background and includes relevant references in my opinion. “Materials and Methods” section is well written and prepares the reader to expect considerations about articles published between 2000 and 2019. But reported papers are older too. Why did you consider them? There are papers of 1987 and 1992! So the paper so written is very confused, too many articles, too many paragraphs and mostly no strong discussion is reported. The reader cannot understand the common thread of the paper, she/he is waiting for strong considerations after a long list of articles. So written, this paper is not ready to be published. In my opinion you should organize the paper in a different way: starting with the list of 19 selected studies, group similar articles and discuss them reporting the common aspects and the differences. So you can write conclusions.

Then, is it possible to obtain information about human exposure? I mean, combined negative effects of these substances on human mothers?

Reviewer 4 Report

Please find my comments below:

The current title "Effects of Environmental Toxicant Exposure on the Maternal Brain and Behaviour: A Scoping Review "needs to be  changed to reflect  either the  potential impact  on the  Canadian  Arctic  region, or the animal-based nature of 95% of the literature.  According to authors,  the most exposed are "populations who consume large amounts of traditional foods from the marine  environment". Line 60: "Although most industrialized countries have banned their production and use, they remain widely dispersed in the environment ". Please provide more specific details about their location in the environment. Line 67: "OCPs are highly persistent in the environment ". Please provide more specific details about their location in the environment. Line 79" "high tropic level species ": do you mean trophic? If yes, correct. Line 80: In "Northern populations ": define and estimate the total number of populations at risk. Lines 103-105: "Exposure to toxic chemicals can result in cellular dysfunction, perturbations in distinct cell populations, and modifications to the plasticity of the mammalian brain ": for every impact provide examples so that a reader not trained in a medical field could understand better. Line 126: "July 11 and 13th 2019. " ensure the same style. Line 140: "exposure to methylmercury, polychlorinated biphenyl congeners, or co-exposure to any of the three toxicants ": since you mention that there are 3 toxicants investigated, but list only two, include the name of the third one as well (OCP). Line 149: "Included studies were reviewed ": specify the total number of studies included. Table 3 lists about 19 studies. Line 153: "Microsoft excel ": make excel in a capital letter as well. In the Results, 1138 studies were considered irrelevant. Explain irrelevancy. Since only one study examined maternal toxicant exposure in humans, it needs to be stated much earlier that literature used in the review was 95% based on animal studies. It should be established early on and also in Conclusions when authors state "This scoping review gives insight into the multitude of effects associated with maternal toxicant exposure" : add here that you used animal-based studies.